# Women Abused: Analysis of Assistance Provided by Urgency Mobile Service

**DOI:** 10.3390/ijerph21010087

**Published:** 2024-01-12

**Authors:** Dalton Makoto Senda, Makcileni Paranho de Souza, Fernando Castilho Pelloso, Raíssa Bocchi Pedroso, Maria Dalva de Barros Carvalho, Sandra Marisa Pelloso

**Affiliations:** 1Postgraduate Program in Health Sciences, Estate University of Maringá, Maringá 87030-230, Brazil; dmsenda@gmail.com (D.M.S.); mpsouza@uem.br (M.P.d.S.); rbpedroso.ct@uem.br (R.B.P.); mdbcarvalho@gmail.com (M.D.d.B.C.); 2Department of Medicine, Federal University of Paraná, Curitiba 80060-000, Brazil; fercaspell@gmail.com

**Keywords:** violence against women, feminicide, pre-hospital emergency care

## Abstract

Considering that reports of violence against women must come after a victim seeks help, the subject matter transcends health-related issues. In Brazil, mobile urgency services (SAMU/SIATE) frequently provide first aid to these women and, to the best of our knowledge, no other research has specifically examined the first reaction given to these women. The present study aimed to analyze SAMU/SIATE assistance to abused women in a cross-sectional study of the assistance to assaulted women provided by SIATE and SAMU Maringá/Norte Novo between 2011 and 2020. Women between 20 and 39 years old, non-pregnant, were the main victims, and 19.52% of them have used drugs of some kind. The (ex) partner figured as the perpetrator in 17.35%, but there was no information about this variable in 73.75% of the records. The Chi-square test shows a mortality rate superior to 70% among the severely traumatized victims. This is the first research work to examine the kind of care that SAMU/SIATE offers, and it identifies several weaknesses in its “*modus operandi*” that may prevent the results from being applied to larger contexts. In addition, further studies on mobile urgent care services in other provinces are required in order to suggest ways to lessen this epidemic.

## 1. Introduction

Violence against women is a health crisis that is considered a silent global epidemic. One in three women all around the world suffer violence, affecting 641 million [1]. Gender-based violence (GBV) refers to any act of violence against a woman causing physical, psychological and sexual affliction, as well as threats or deprivation of liberty. This encompasses a series of forms of violence that affect women and girls, including child abuse, sexual violence, and domestic violence, in addition to its most serious and extreme form, feminicide (women killed by gender violence). Aggression or violence by an intimate partner (IPV) is the most common form of GBV [2].

Several studies indicate that the most common type of violence suffered by women around the world is psychological, followed by physical and sexual violence [3,4,5,6].

In the Americas and the Caribbean, the prevalence ranges from 1% in Canada to 27% in Bolivia [7], corroborating the estimated prevalence of 29.8% made by the World Health Organization [1].

In Brazil, approximately 43% of women do not feel completely respected by their partners [8], and it is one of the nations with the highest rates of feminicide all around the world. According to data from the United Nations Economic Commission for Latin America and the Caribbean [9], 40% of all feminicides in both regions occur in Brazil, which ranks the country in 5th place for feminicides among 84 countries in the world [10].

In Brazil, the total number of crimes against women and feminicides increased every year [11], despite the existence of laws intended to prevent gender-based violence. Act No. 10.778/2003 made the reporting of violence against women mandatory, starting in 2003. The Feminicides Act (Act No. 13.104/2015), also known as the Maria da Penha Act (Act No. 11.340/2006), and its stricter penalties, are two further essential initiatives in the campaign against and prevention of this crime [12].

“One of the Millennium Goals” was launched by the World Health Organization as part of the 2030 Agenda to completely eliminate all forms of discrimination against women and girls worldwide. Global efforts are still required, though, to put into practice measures that will strengthen the prohibition against discrimination in the workplace, promote gender equality in public life, marriage, and family, enact stronger legislation, and end all forms of violence in both public and private settings, including human trafficking and sexual exploitation [13].

The issue of violence against women is becoming more and more prominent, as seen by the rise in the number of debates to understand its conditions, whether in domestic settings among friends and family or in the media, law, or politics [14,15,16]. Knowing how common violence is in a certain area might help with the creation of new public policies and initiatives meant to reduce violent crimes and, consequently, the startlingly high rate of feminicides.

However, the topic of violence against women is not limited to the medical field. When victims seek attention from medical services, the incidents are reported [17]. In Brazil, mobile emergency services (SAMU/SIATE) frequently offer these services. Interestingly, there are not any studies that provide precise information about the support that these mobile services give to women who have become victims of aggression. Our study intends to extensively investigate cases of aggression experienced by women and handled by mobile emergency services (SAMU/SIATE), considering the significant gap in the literature.

By concentrating on this particular aspect, our research hopes to provide a deeper comprehension of the experience of women who are victims of violence, as well as insight into the effectiveness of emergency services in handling and resolving these instances. Our research also aims to pinpoint possible areas where the current response systems could be strengthened and to provide guidance for the creation of focused initiatives that would better assist and safeguard women in certain violent circumstances.

## 2. Materials and Methods

This is a cross-sectional study of aggression against women cases conducted by the Emergency Care Service (SIATE and SAMU Maringá and Norte Novo) from 2011 to 2020. The primary objective of this research is to comprehensively analyze incidents of aggression against women and understand various aspects related to these cases.

The study is set in the municipality of Maringá, situated in the southern region of Brazil, which ranks as the third most populous city in the State of Paraná, with an estimated population of 436,472 inhabitants [18]. The population makeup is predominantly white (71%), followed by mixed race (22%), Asian (4%), Black (3%), and less than 1% Indigenous [19].

### 2.1. Data Source

The main source of the data was the SAMU Norte Novo regulation center, and data were collected between September 2020 and August 2021. Data were obtained directly from the physical files of the Advanced Mobile Support Units’ Medical Care Records, Nursing Care Records, and Assistance Registry (Portuguese Acronyms: RAM/RAE) and Basic Mobile Support Units’ registrations (Portuguese Acronym: RAS).

The study covered all occurrences of aggression against female victims that were determined to be such, and Microsoft Excel^®^ spreadsheets were used to systematically gather the data. The researcher carefully reviewed and examined each record on an individual basis in order to reduce bias and guarantee data consistency.

### 2.2. Variables

The investigation concentrated on a number of important factors, such as the victims’ ages, the aggressors’ traits, whether or not the victims were pregnant, the intensity of the trauma, drug usage, and outcomes like death. The common variable, aggression, was then connected with each of these independent variables. The day of the week and the time of the occurrences are described as supplementary information.

### 2.3. Data Evaluation and Statistical Analysis

The individual analysis of the variables was subjected to descriptive statistics and represented graphically. Based on the indicators of the locations where the attacks occurred, grouped by neighborhood and municipality, heat graphs were generated, pointing to the regions with the highest records of violence. All the analyses were performed with the aid of the statistical environment R (R Development Core Team), version 3.5 [20], adopting a 5% level of significance (*p* value < 0.05).

The types of aggression were characterized according to the International Classification of Diseases (ICD-10), using the codes Y09 (Unspecified Aggression), X93 (Assault by Handgun Discharge), X99 (Assault by Sharp Object), and T74.2 (Sexual Abuse). Due to the low number of observations within each category, the results generated by the regression were not significant, and therefore these variables were regrouped in a dichotomous way, creating an indicator variable named “Nature of aggression” that includes:

Nature of aggression: “Non-Specified” (*n* = 332);

Nature of aggression: “Specified” (*n* = 129).

Assaults of a specified nature included codes X93, X99, and T74.2. The other categories of physical aggression, as they did not have specific coding (ICD-10), were considered to be of an unspecified nature (code Y09). With that, it was possible to make use of the binomial family logistic regression model, increasing its predictive capacity.

The equation for the applied logistic regression model is written as:(1)$$p=\frac{1}{1+e^{-b_{0}+b_{1}x_{1}+...+b_{k}x_{k}}}$$
where $b_{k}$ are the regression coefficients and $x_{k}$ are the explanatory variables.

Also, we analyzed and discussed the incidence of occurrences regarding the day of the week and time.

The spatial distribution of the occurrences is illustrated through heat maps.

### 2.4. Ethical Aspects

The State University of Maringá Ethics Committee Involving Human Beings, protocol No. 3.071.844, approved this project.

## 3. Results

Data on violence, whether regional, municipal, national, or international, must be analyzed in order to understand its causes and the risks associated with the act, the victim, and the aggressor. Urgency and emergency services are qualified to carry out this type of assistance. Between 2011 and 31 December 2020, SIATE and SAMU Norte Novo recorded 461 incidents of aggression against women.

### 3.1. Sample Characteristics

The age group with the highest occurrence ranged between 20 and 39 years old, with a median of 31 years old, gradually decreasing from then on, until reaching a maximum age of 87 years old. It was found that 5.21% of the victims were pregnant, 78.52% were not pregnant, and 16.27% of the records did not contain this information.

Regarding drug use, 19.52% of women used some type of drug, with alcohol being the most common; the other 80.48% did not declare this information. Records with information on the use of illicit drugs or a combination of alcohol + illicit drugs were grouped into the use of “illicit drugs”, with 2.6% use of other drugs, without mentioning alcohol.

Physical aggression covers all (100%) of the events, being the only variable common to all records. In correlating the nature of physical aggression (dichotomous distribution) with the other variables, the most common was the “Non-Specified” (NS aggression), with 72.02% of cases. Among the “Specified” attacks, 11.5% were stab wounds (SW), 10.2% were firearm injuries (FAI), and 4.56% were victims of sexual violence (SV) (Table 1).

When associating the severity of trauma with the outcome “Death” among victims who suffered trauma considered severe and measured by the GCS (Glasgow Coma Scale), 71% died. The application of the chi-square test indicates the dependence between these variables, which means that the severity of the injury caused by the aggressor impacts directly on a woman’s death (Table 2). We observed that 10.24% of victims refused referral after assistance from SAMU professionals and 4.52% of records did not mention referrals. The chi-square test (*p* = 0) demonstrated dependence between the severity and referrals (Table 3).

In 73.75% of the records there was no information about the aggressor. In 17.35% of cases, the partner (husband/boyfriend) appeared as the aggressor, and in lower percentages, other perpetrators appeared, such as parents or stepfathers, children, siblings, other family members, and acquaintances (Table 4).

The variable day of the week and time of aggression demonstrated that a percentage of 58.14% of aggressions occurred between Monday and Friday (*n* = 268), with a higher prevalence at night (39.7%). Then, the dawn and afternoon periods presented a very similar distribution of occurrences, around 23.4% each. In the morning there were fewer cases attended, comprising 13.45% of records, but there was still a high incidence (*n* = 62).

We observed that the type of aggressor, severity of the injury, and death had a statistically significant association with the nature of the aggression. The aggressors were mostly (ex) partners and family members, and committed violence by unspecified means, which resulted in less severe injuries. The most serious injuries, including those resulting in death, resulted from injuries caused by firearms and bladed weapons. In this study there was no significant correlation between the variables pregnancy and drug use and the nature of the attacks.

In the correlation regression model between the identified nature of the attacks (dichotomous distribution) and their incidence in different locations and different age groups, we did not find significant values. Geographic location also did not influence the nature of the attacks.

The severity of the aggression was measured using the Glasgow Coma Scale (GCS) found in the case forms. Mild trauma (GCS between 13 and 15) was the vast majority (91.11%) and severe trauma (GCS between 3 and 8) accounted for 6.72% of cases. Trauma considered moderate (ECG between 9 and 12) corresponded to 0.43%, and 1.74% (*n* = 8) of records did not have any information on severity according to the GCS.

### 3.2. Spatial Distribution

Heat maps were generated in order to facilitate a visual understanding of the geographic points of occurrences. For its construction, the addresses contained in the RAS and RAM/RAE were used, which originated the coordinates from the neighborhood where the aggression was recorded.

The Figure 1 llustrates all cases in the period analyzed in Maringa and the region:

Most hotspots were located in the central region of the city of Maringá, and other hot spots occurred in neighboring regions.

The Figure 2 shows, in more detail, only the Maringá occurrences during the entire study period. There is a large accumulation of cases occurring in central regions of the city and extending across the Zone 07 neighborhood, including Morangueira Avenue and Jardim Alvorada neighborhood. Some other places with lesser occurrences were found northwest and south of Maringá.

The distribution of aggression incidents against women proved to be quite heterogeneous, with records in all regions of the municipality. However, throughout the period studied, it is consistently observed that there was a higher prevalence of violent actions against women in the central region, Zone 07, and Jardim Alvorada neighborhoods.

## 4. Discussion

This is the first study to analyze cases of aggression suffered by women treated by mobile emergency services (SAMU/SIATE). In a study carried out in Italy, with data from an Emergency Service during 2017 to 2020, the authors observed an increase in the rate of domestic violence in the first wave of the pandemic [21]. We observed a non-significant decrease in records; however, violence is often underreported. Evidence suggests that the COVID-19 outbreak has also reduced access as well as help-seeking by victims [22,23]. People who needed SAMU’s care were satisfied with its resolutivity, demonstrating the importance of this service for immediate assistance in urgent and emergency cases [24].

Studies have demonstrated the significance of examining characteristics related to violence from mobile care services, in order to enhance epidemiological data, empower practitioners to implement best practices, and enhance access to healthcare services. Emergency services personnel frequently deal with unpredictable acts because, in most cases of violence, there is a confrontation between the victim, the aggressor, and the police, whose training usually does not prepare them for the complexities of such a report. When health teams in pre-hospital services are faced with a case of violence against a woman, they should prioritize her physical safety, give her medical attention and treatment, show compassion and support, document the findings, and report the incident, according to a study conducted with these teams [25].

The main reason for assistance was physical and/or sexual aggression. Despite being a silent epidemic, physical violence has several particularities that contribute to its occurrence, such as unemployment, the abuse of legal and illicit drugs, and low levels of education [26]. The major factors linked to violence include race and gender, despite substantial debate on the subject. Black women aged between 20 and 39 are the most affected [27,28]. This study brought forward important information about women who suffered aggression and whose data are not part of the system’s statistics, since continuity of care does not always occur.

The results show that women aged between 20 and 39 are more exposed to physical aggression. This group represented more than half of the women attacked throughout the period; however, the data are insufficient on the races of the victims. Recently, a study found an even higher prevalence of facial trauma in women victims of GBV in this age group [29].

Concerning the women victims of sexual violence (SV), the majority did not identify the aggressor. Data from DATASUS, however, indicates that the vast majority of perpetrators of sexual crimes are known people or family members. Strangers are responsible for approximately 22% of these crimes [30]. This omission may be associated with the fear of identifying someone close, which can lead to reprisal and oppression from the perpretator.

Among the abused women who sought assistance from the mobile services, and who deserve greater attention due to their particularities, are pregnant women. Although this study identified only a small percentage, it deserves to be investigated, given that the statistic should be zero. Several researchers, however, report prevalence rates of assaulted pregnant women ranging from 1.2% to 27.6% in some countries [31]. Violence against pregnant women is an act that not only affects the health of the abused woman, but also the fetus in her womb, which may not be born at all, be born prematurely, and/or have a low birth weight [32].

Among the findings of this study, the use of some type of drug was verified, but there was no correlation between this variable and the severity or nature of the attacks. It is known, however, that the use/abuse of legal and/or illicit drugs, whether by the victim or the aggressor, is another factor that appears to be associated with the occurrence of violent acts [33,34].

The majority of women were victims of assault by means of physical force/beating (ICD-10: Y09), followed by injuries by means of a cutting or penetrating weapon (ICD-10: X99), firearm shooting (ICD-10: X93), and sexual violence (ICD-10: T74.2). Women who were victims of burglary were included in the group of assaults involving physical force. The observation of violent acts associated with being a woman is increasingly frequent in our society, a fact that is directly linked to patriarchal cultural aspects. A study comparing the description of intimate partner violence, from the perspective of men and women, shows that men justify their acts of aggression through vague speeches, with euphemisms, often blaming a third party for the inflicted act [35].

An analysis of the profile of IPV cases treated in urgency and emergency services in 25 Brazilian capitals revealed very similar numbers regarding drug use (21.98%) and the prevalence of means of aggression, with 70.9% for bodily force/beating and 14.5% for injuries caused by a sharp or penetrating weapon [27]. In this study, however, the prevalence of victims due to firearm shooting was much lower than that found in the Maringá region. Part of this difference can perhaps be explained by the environment analyzed, since in the study of those authors, hospital care was evaluated and, in our study, we evaluated pre-hospital care. We found 47 records of aggression by firearms, and 33 were referred to a hospital service; the other victims were either sent directly to the Forensic Medicine Institute (FMI) or refused referral.

Considering the severity of the injuries, a few cases were classified as severe, according to the Glasgow Coma Scale. Tragically, however, a high rate of death was observed among these cases. There were 22 deaths among 31 victims with severe trauma, which represents almost 71% fatality. This percentage surpasses the lethality rate of severe or congestive heart failure, which is 50% in 5 years [36] and, in Brazil, cardiovascular diseases are the main cause of death [37].

IPV is the most common form of violence against women [1]. The database utilized in this study contains little information about the perpetrators of the attacks and, when present, it was handwritten in the body of the anamnesis text. This lack of information is possibly associated with the health professional approach towards the assaulted woman, by not asking the victim and/or witnesses this important information. Some factors in the context of emergency care are considered to explain the failure to register, such as the in-depth care of cases, rapid intervention, flow of care, disarticulation with the reference service, eagerness to resolve the problem, and lack of preparation to approach and deal with the situation [38].

One further relevant factor is the time frame during which the violence took place. This hostility can erupt at any moment and in any place. Research shows that on weekdays and at night, the prevalence is greater [29,39], while others point to weekends as the highest risk days for women [40,41]. Women in our study had fewer insults on weekends and greater aggression at night, although many of them experienced physical violence at all hours of the day.

The domestic environment should be women’s last refuge, her home. But IPV in their own home remains the most common form of aggression suffered by women [28,42]. Factors such as low income, low level of education, and living in rural areas are associated with a higher prevalence of violence against women [43]. In the present study, the information contained does not detail the locations of the occurrences. But what was noticed through the heat maps was the predominance of events in the urban area, and mainly in the most central regions of the city of Maringá.

Maringá is a city in the southern region of Brazil, with a predominantly urban population (>98%) and with a Human Development Index (HDI) considered very high (0.808) [18]. The database does not include sociodemographic aspects, with evidence of a higher prevalence in the most populous and demographically dense regions of the city (central region, Zone 07, and Jardim Alvorada) [18]. This may be viewed as merely reflecting what goes on in places with higher population densities, placed inside a setting with a more consistent socioeconomic environment.

The study examined cases of aggression against women treated by mobile emergency services but acknowledged limitations due to the secondary nature of the data collection and the subjective registration process. This resulted in gaps and missing information, especially in cases of aggression against women. Crucial details like location, perpetrator identification, and victim’s race were often absent, highlighting a lack of training by health professionals to document such cases adequately. The database lacked sociodemographic information, limiting a comprehensive analysis of the socioeconomic context and environmental factors contributing to violence against women. A more detailed investigation of sociodemographic influences was not possible, despite the study’s indications that services are concentrated in Maringá’s central regions and urban areas. Additionally, the study’s focus on a specific region, Maringá, limits the export of the findings to other contexts. Variations in healthcare practices, cultural dynamics, and socioeconomic conditions in different regions could influence the patterns and documentation of cases.

To address these limitations, future research endeavors could benefit from exploring mobile care services in diverse geographic locations and involving a broader spectrum of sociodemographic factors, finally allowing the better development of more robust strategies to combat gender-based violence.

## 5. Conclusions

Society and governments can no longer tolerate acts of violence against women. The consequences of this crime add up to an endless number of afflictions, with negative repercussions in health, education, and the economy, while also representing an important brake on the evolution of nations, and a cursed inheritance passed down along generations.

Aiming to protect precisely the most exposed population, it is argued that prevention measures against GBV should begin before the first marriage and the age of 19 [44].

In Brazil, despite the existence of measures and laws to restrict and strictly punish this crime, there is a progressive increase in cases. This portrays a structural problem, rooted in the culture and habits of a patriarchal society. Therefore, the fight against this issue, like many diseases, must focus on prevention, which should involve access to information and changing customs and paradigms accepted since childhood. The seeds of tolerance and the notion of equality, including gender equality, must be planted in elementary education, becoming a formal part of the curriculum.

Currently, research in the field of gender-based violence has made significant strides in understanding the prevalence and patterns of violence against women. Existing studies have highlighted the impact of socioeconomic factors on the occurrence of violence and have identified key risk factors such as age and race. However, there is a notable gap in the literature concerning the documentation and analysis of cases treated by mobile emergency services. This study seeks to address this gap by providing a comprehensive analysis of incidents recorded in such services, offering insights into the nuances of violence against women in emergency care settings. While previous research has largely focused on post-incident outcomes, our study aims to shed light on the immediate responses and challenges faced by healthcare professionals in these critical situations.

Therefore, in view of what was observed in this study and considering not only the importance of mobile assistance services as a damage control instrument, but also their potential in the secondary prevention of these crimes, we propose that SAMU/SIATE adopt adequate training, and the use of a specific assistance form for incidents of aggression against women, with data that allow the identification of risk factors and also of aggressors.

## Figures and Tables

**Figure 1 ijerph-21-00087-f001:**
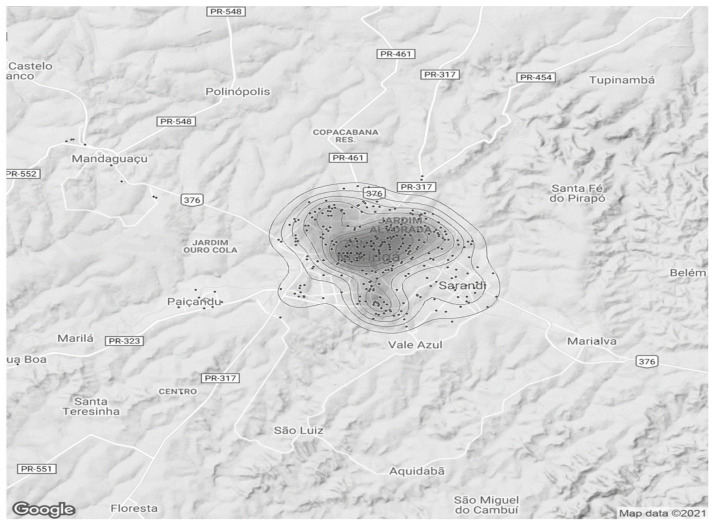
Map 1: Aggressions against women between 2011 and 2020.

**Figure 2 ijerph-21-00087-f002:**
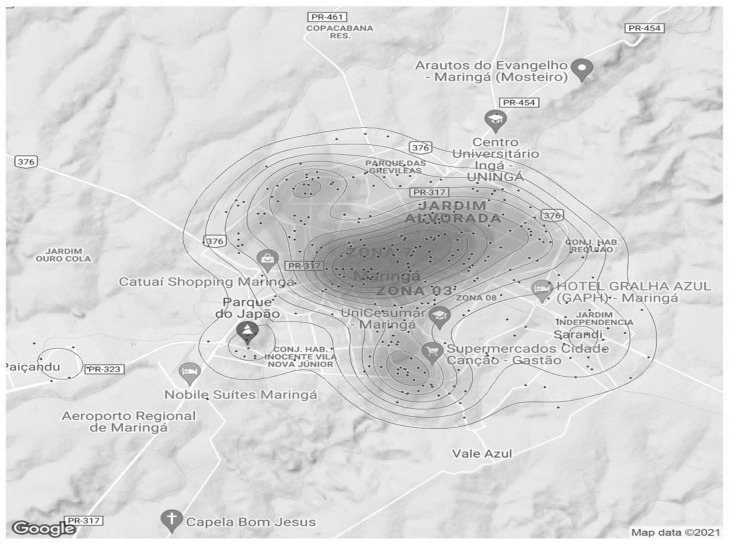
Map 2: Aggressions against women by neighborhood from 2011 to 2020.

**Table 1 ijerph-21-00087-t001:** Characterization and description of victims treated by the Mobile Emergency Service. Maringá, PR, 2022.

Variable	Nature of Aggression				*p*-Value (*X*^2^)	Odds Ratio
	Non-Specified (*N* = 332)		Specified (*N* = 129)			
	*n*	%	*n*	%		
Aggressor						
Partner	75	22.59%	5	3.88%	<0.0001	7.24
Acquaintence	6	1.81%	0	0.00%		Inf
Ex-Partner	4	1.20%	1	0.78%		1.56
Family Member	27	8.13%	3	2.33%		3.72
Uninformed	220	66.27%	120	93.01%		0.15
Pregnancy						
No	255	76.81%	107	82.95%	0.077	0.68
Uninformed	55	16.57%	20	15.50%		1.08
Yes	22	6.63%	2	1.55%		4.51
Severity						
Severe	7	2.1%	24	18.6%	<0.0001	0.09
Mild	319	96.09%	101	78.3%		8.42
Moderate	2	0.61%	0	0.00%		Inf
Uninformed	4	1.2%	4	3.1%		Inf
Age						
<20	46	13.86%	26	20.15%	0.26	0.63
>60	19	5.72%	4	3.1%		1.87
20–39	178	53.61%	67	51.94%		1.06
40–59	89	26.81%	32	24.81%		1.14
Death						
No	329	99.10%	110	85.27%	<0.0001	18.94
Yes	3	0.90%	19	14.73%		0.05
Drug Use						
Illicit drugs	9	2.71%	5	3.88%	0.074	0.69
Licit drugs	54	16.27%	10	7.75%		2.31
Uninformed	259	78.01%	112	86.82%		0.54
Others	10	3.01%	2	1.55%		1.97

**Table 2 ijerph-21-00087-t002:** Deaths based on the severity of the injury.

Severity	Death	Amount	Percentage
Severe	No	9	29.03%
Severe	Yes	22	70.97%
Mild	No	420	100%
Moderate	No	2	100%

**Table 3 ijerph-21-00087-t003:** Referral based on the severity of the injury.

Severity	Referral	Amount	Percentual
Severe	Medical referral	9	29.03%
Severe	Forensic Medicine Institute (FMI)	22	70.97%
Mild	Medical referral	358	85.24%
Mild	Uninformed	19	4.52%
Mild	Refused referral	43	10.24%
Moderate	Medical referral	2	100%

**Table 4 ijerph-21-00087-t004:** Aggressor identity.

	Amount	Percentual
Partner	80	17.35%
Acquaintence	6	1.3%
Ex-Partner	5	1.08%
Family member	30	6.51%
Uninformed	340	73.75%

## Data Availability

No new data were created or analyzed in this review. Data sharing does not apply to this article.

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
