# Peer review of "Women Abused: Analysis of Assistance Provided by Urgency Mobile Service"

_ijerph, 2024, doi:10.3390/ijerph21010087_

Round 1
Reviewer 1 Report
Comments and Suggestions for Authors
Authors can improve the references and discuss about the the impact of COVID-19
Author Response
Replies in the attachment file. Thank you.

Reviewer 2 Report
Comments and Suggestions for Authors
Questions/Suggestions
Q1. The Qui- 18 square test shows a mortality rate superior to 70% among the severe traumatized victims – in comparison with non-victimized women?
Q2. It is strongly suggested the adoption of the term femicide or feminicide, once it is the most extreme form of VGB. A definition should be presented and the concept uniformized - e.g., along the manuscript, the authors refer to female homicide, femicide and feminicide without presenting the (genderized) concept, even though is strongly associated with research core.
Q3. Violence against women is a problem that goes beyond the health sphere, since after the victim seeks care at a health service, the act is reported [18] – Even being reported there is no information about who’s the perpetrator? Wasn’t it possible to cross information with authorities reports?
Q4. Since 2003 the violence against women became an item of compulsory notification (Act n° 48 10.778/2003), other important measures to fight against and prevent this crime are the 49 Maria da Penha Act (Act n° 11.340/2006) and the Act nº 13.104/2015, also known as the 50 Feminicide Act, which toughened the penalty for this crime. It is suggested that a brief description of these laws is made, otherwise it is pointless to have this information in an reader’s perspective.
Q5. (…) and, as far as we know, there are no studies with specific data on the assistances provided to women victims of aggression, carried out by these mobile services. This type of affirmation should be avoided, whether there is or there isn’t (published) data?
Q6. We strongly doubt that the characterization of the city population in terms of ethnicity is important to the study. We strongly suggested to retrieve the expression “race” as it is scientifically dated (e.g., 39; 245). It is suggested the adoption of “racialized victims” in order to address the added vulnerabilities of these women (i.e., sexism, racism).
Q7. Regarding to the method, was it a team of one? Were the variables on the grid discussed with other (supervisor?). If so, it should be mentioned.
Q8. Why the variables chosen these and not others? age of the victims, aggressor, pregnancy, trauma severity, drug use and death. This process should be cleared and empirically sustained. Additional information regarding the research option to take drug use as a variable. Once there is no information regarding the perpetrator in most of the events, the information only regarding the victim can promote biased interpretation e.g., victim blaming. On the other hand, the victim could had used drugs in an self-medication strategy post attack.
Q9. It is suggested that a mix method methodology is presented, concretely document analysis (?) and retrospective epidemiological. What does the research team understand by observational? (72)
Q10. Why the time frame is 2011 and December 31, 2020? The research option should be presented.
Q11. Regarding with violence severity, how did the research team define severe, mild and moderate? Was it adopted Glasgow Coma Scale (GCS)? It should be presented a brief resume of the scale variables. For example, if a victim of sexual assault is 100% responsiveness: eye-opening, motor, and verbal responses, it means that the violence was moderate? If this is true, it can bias the results interpretation.
Q12. To discuss these results under GBV frame, it would be extremely important to know who the perpetrator was, at least if known or not. Once the data regarding domestic violence, violence towards women that are assaulted by strangers, women that are caught in thirds situation (e.g., open fire) are mixed, with less than 1/5 of this information present, it facilitates biased interpretation.
Q13. Data regarding day of the week and time are presented for the first time on Results. The Spatial distribution as well.
Q14. You don’t know if it is the first study? Or it is the first published study? (217)
Comments on the Quality of English LanguageThe English presented is non technical with misspelling detected and informal/poor sentence construction.
Author Response
Replies in the attachment file.

Reviewer 3 Report
Comments and Suggestions for Authors
Thank you for inviting to review this manuscript. This is an interesting paper that sheds light on the assistance of SAMU/SIATE mobile Services to women suffering gender-based violence. However, as it stands, the paper needs to address a number of issues to achieve publishable standard. These are explained below:
First, the structure related to the literature section is hard to follow and for this reason the paper seems to lose overall coherence of argument. Authors need to explicitly state what they will do in the study, how their study will contribute to the current literature, what research question they will ask or what hypothesis they will make.
Second, some demographic facts would be of interest to the reader to be able to understand the result.
Third, I don’t feel that there is enough discussion of the specific nature of how the data was analysed to be able to determine the authenticity or trustworthiness of the findings.
Comments on the Quality of English LanguageEnglish needs editting. There are several grammar and ortographic mistakes throughout the paper.
Author Response

(The authors gave the same response as above.)

Round 2
Reviewer 2 Report
Comments and Suggestions for Authors
Although the authors replied to every question made, even though some remit to "internet era", it does not adress the methodological limitations of the study. We understand that the limitations are partially due to the system itself, but in that note, the methodological options, concretely the associations made and the theoretical matrix should be more robust.
Comments on the Quality of English LanguageThe english present is commonly informal and not coherent to scientific publication.
Author Response
Thank you for your time evaluating our manuscript. The responses to the reviewers are uploaded above.

Reviewer 3 Report
Comments and Suggestions for Authors
I congratulate the authors on their effort to improve and adequate their manuscript to the comments and suggestions made. However, there are several issues that need to be further addressed before publication. Please find my comments below:
1. The structure related to the literature section of the document is still hard to follow. In this sense, I suggest authors reorganize the information already written in order to make it coherent.
2. In line with the previuos suggestion, the authors need to explicitly state what they will do in the study, how their study will contribute to the current literature, what research question they will ask or what hypothesis they will make.
3. The authors also need to give a brief description of what is the current state of research in this area.
Author Response
Thank you for the time spent evaluating our manuscript. All the responses to the reviewer are in the document below.
